# Understanding Melt Pool Behavior of 316L Stainless Steel in Laser Powder Bed Fusion Additive Manufacturing

**DOI:** 10.3390/mi15020170

**Published:** 2024-01-23

**Authors:** Zilong Zhang, Tianyu Zhang, Can Sun, Sivaji Karna, Lang Yuan

**Affiliations:** Department of Mechanical Engineering, University of South Carolina, Columbia, SC 29201, USA; zilongz@email.sc.edu (Z.Z.); tz5@email.sc.edu (T.Z.); cans@email.sc.edu (C.S.); skarna@email.sc.edu (S.K.)

**Keywords:** additive manufacturing, laser powder bed fusion, fluid dynamics, melt pool instability, surface topography

## Abstract

In the laser powder bed fusion additive manufacturing process, the quality of fabrications is intricately tied to the laser–matter interaction, specifically the formation of the melt pool. This study experimentally examined the intricacies of melt pool characteristics and surface topography across diverse laser powers and speeds via single-track laser scanning on a bare plate and powder bed for 316L stainless steel. The results reveal that the presence of a powder layer amplifies melt pool instability and worsens irregularities due to increased laser absorption and the introduction of uneven mass from the powder. To provide a comprehensive understanding of melt pool dynamics, a high-fidelity computational model encompassing fluid dynamics, heat transfer, vaporization, and solidification was developed. It was validated against the measured melt pool dimensions and morphology, effectively predicting conduction and keyholing modes with irregular surface features. Particularly, the model explained the forming mechanisms of a defective morphology, termed swell-undercut, at high power and speed conditions, detailing the roles of recoil pressure and liquid refilling. As an application, multiple-track simulations replicate the surface features on cubic samples under two distinct process conditions, showcasing the potential of the laser–matter interaction model for process optimization.

## 1. Introduction

Laser powder bed fusion (LPBF) has emerged as a pivotal technology in the realm of additive manufacturing (AM) for the production of high-performance metallic components. This technique provides a distinctive capability to fabricate intricate geometries with exceptional resolution, customize the mechanical properties of components, and minimize lead time while reducing material waste [1,2,3,4,5]. It finds applications across diverse industries, such as aerospace, automotive, healthcare, and tooling, where high precision and specific material properties are crucial. Notwithstanding its advantages, LPBF presents challenges, such as the need to optimize process parameters to achieve the desired part quality and address defects during the printing process [6,7,8,9,10]; the laser–matter interaction in LPBF creates the melt pool, the fundamental element forming the desired geometrical design track by track and layer by layer and determining the quality of the builds. Insufficient melts or excessive heat input can lead to lack-of-fusion or keyholing defects [11,12,13,14,15]. The high instability introduced due to the fluid dynamics in the melt pool can cause spatters, balling, or humps that potentially affect the integrity of the materials [16,17,18,19]. Therefore, understanding the laser–matter interaction at the melt pool scale and its impact on surface characteristics is critical to assist in the development of the process.

In situ technologies through high-speed imaging, e.g., using X-ray radiography and optical or thermal cameras, have been employed exclusively to investigate the melt pool dynamics and understand the laser–matter interaction [20,21,22,23,24,25,26]. Zhao et al. [27] probed the dynamics of the Ti-6Al-4V in the LPBF process from the inside, as well as above the surface of, the powder bed using high-speed hard X-ray imaging and diffraction techniques, concluding that the competition between the Marangoni convection and the recoil pressure primarily determines the extent of the mass, the heat transfers, and the dynamics of the melting process. Yin et al. [28] investigated backward-ejected spatters of IN718 using a high-speed camera in the LPBF process. They found that the expanding vapor plume mainly drove the ejected spatters. Guo et al. [29] reported directly observing and quantifying the melt pool variation of AlSi10Mg via in situ high-speed high-energy X-ray imaging. They pointed out that the melt pool can undergo different melting regimes, and both the melt pool dimension and volume can change orders of magnitude under a constant input energy density. Leung et al. [30] revealed the underlying physical phenomena of Invar 36 during the deposition of the first- and second-layer melt tracks through in situ and operando high-speed synchrotron X-ray imaging. They uncovered mechanisms of pore migration by Marangoni-driven flow, pore dissolution, and dispersion by laser remelting. The vapor plume flew backward, impacted the free surface, especially the melt pool end region, and formed backward spatters. As an observation, both the recoil pressure and vapor plume play a critical role in driving melt pool behavior during laser scanning. 

Considering the limits of in situ characterization techniques in LPBF involving limited spatial resolution, difficulties in capturing complex three-dimensional structures, and high cost, computational models that simulate the melt pool dynamics by using Computational Fluid Dynamics (CFD) to account for the fluid flow, heat transfer, and phase change within the melt pool have been applied to interpret the experimental data and assist the process development [31,32,33,34,35,36,37,38,39]. During this process, physical forces, including surface tension, viscous forces, recoil pressure, and gravity, interact within the molten pool and influence material flow and heat transfer [16,40,41,42,43,44,45]. Tang et al. [16] simulated 316 L stainless steel tracks in LPBF. Their single-factor analysis studied the effect of surface tension, Marangoni shear force, viscous force, and recoil pressure on the humping phenomenon in laser powder bed fusion, respectively. The results indicated that recoil pressure played a more critical role than the other three forces in the melt pool. Ren et al. [46] numerically explored the influence of the contour scan sequence on the surface roughness magnitude in the LPBF of copper alloy using a high-fidelity CFD model. They found that the presence of powders deposited on the infill region helped enhance the laser absorption during the pre-contour scan and, by extension, improved the melting behavior and surface flatness. Li et al. [47] developed a particle-scale CFD method with three lasers in LPBF. The molten track of the leading laser randomly deviated from one of the molten tracks of the auxiliary lasers due to the uneven distribution of the powder bed, which caused balling and pits, leading to the extreme degradation of surface quality. Dai et al. [48] suggested that excessive recoil force can cause powder spatters and keyholes, while the inefficient Marangoni effect led to the non-uniform melting of IN718 in LPBF with a high-fidelity CFD model. Prior research demonstrated the ability of multi-physical models to accurately predict the quality of melted materials and contribute to the development of new materials. 

One of the direct impacts of the solidified melt tracks is the surface topography and roughness resulting from melt pool morphologies. In LPBF, the surface of a part forms through the layer-by-layer deposition of melted powder material, which melts and solidifies rapidly upon exposure to laser energy, resulting in the final surface characteristics of the printed component. The process parameters, such as laser power and scanning speed, powder characteristics, and scan strategies, have been studied to explain the surface characteristics by incorporating the interaction between the laser and powder bed [1,49,50,51,52]. For example, Guo et al. [53] observed the influence of laser power, scan speed, and hatch spacing on the cube top surface roughness of IN738LC during the LPBF process. Higher laser energy density with lower laser speed leads to complete powder melting, aiding in creating a flat surface. Qiu et al. [54] found that increased laser speed and thicker layers result in irregularly shaped laser tracks and a higher occurrence of discontinuities on the top surfaces for Ti-6Al-4V samples. Calignano et al. [55] concluded that the scanning speed has the most influence on surface roughness for AlSi10Mg due to the introduced high melt pool instability. Yang et al. [56] reached similar conclusions for Ti-6Al-4V, where scanning speed dominated the instability, followed by laser power and layer thickness. 

Experiments and computational modeling in LPBF constitute an integrated approach essential for understanding and optimizing the additive manufacturing process. The synergy between empirical observations and computational simulations aids in advancing the fundamental knowledge of LPBF [57]. This study performed single-track experiments with a wide range of laser power and speed under LPBF conditions to examine the variety of melt pool morphologies. A high-fidelity computational model, based on the commercial software ANSYS Fluent (version 2022 R1), was developed to interpret the experimental results and investigate the underlying mechanisms that lead to dramatically different melt pool behaviors. Then, multiple-track simulations were performed to further assist in explaining surface roughness and correlate the surface topography with the melt pool behaviors, providing new insights into understanding the as-printed top surfaces in LPBF.

## 2. Experimental and Modeling Methods

### 2.1. Experimental Procedure

Experiments based on single lines and cubes in LPBF were performed to understand the melt pool behavior and provide data to validate the computational model. The LPBF machine, Aconity MIDI (Germany), which equips a single-mode fiber laser with a maximum laser of 1000 W, was utilized to print those samples in the argon environment. During the printing, the oxygen level in the chamber was controlled to be lower than 100 ppm to minimize oxidation. The gas-atomized powder was purchased from the Carpenter Technology Corporation (USA). It has a particle size distribution from 15 to 45 μm with a mean size of 30 μm. The laser spot size was kept constant at 100 μm.

Single-track experiments with and without a layer of powder were performed to evaluate the melt pool dynamics under a variety of process conditions. In the later discussion, the terms ‘bare plate’ and ‘powder plate’ are used to represent the conditions without and with powder, respectively. With the powder plate, the powder layer thickness was set at 50 µm. Both the bare plate and the powder are made of 316L stainless steel. The experiment design is shown in Table 1. The cases can be grouped into three series: constant laser power ($P_{laser}$ = 260 W), constant laser speed ($V_{laser}$ = 1.47 m/s), and constant laser energy density (300 J/m). Note that the linear energy density is used for the single tracks, which is defined by $\frac{P_{laser}}{V_{laser}}$. Each line was printed twice with a length of 20 mm. 

To further understand the melt pool dynamics in relation to the top surface roughness, cube samples with the size of 10 mm × 10 mm × 10 mm were printed with low laser power and speed (260 W and 0.52 m/s) and high laser power and speed (440 W and 1.47 m/s). For the cube samples, a simple hatching strategy, where the laser runs back and forth across the entire sample, was applied with a 90° rotation angle between the consecutive layers. The hatch spacing and the layer thickness were set at 100 μm and 30 μm, respectively. 

The Keyence VHX-5000 digital optical microscope (Itasca, IL, USA) was used to obtain the top surface topography with height information for all printed samples, including the single lines and the cubes. To examine the melt pool depth and the detailed topography of the cross sections, the lines were sectioned in the middle of the line, perpendicular to the laser scanning direction, using wire-EDM. To reveal the melt pool boundaries, all samples were polished and etched with a mixture of 75 vol% HCl and 25 vol% HNO_3_. Both the melt pool depth and width were measured to quantify the melt pool dimensions. 

### 2.2. Modeling Method

#### 2.2.1. Governing Equations 

To simulate the melt flow dynamics during LPBF, a CFD model was developed using the commercial software ANSYS Fluent. Assuming Newtonian flow behavior, the solved key equations are provided below [46,58]. 

As part of the Navier–Stokes equations, the conservation of mass is described as:(1)$$\frac{\partial \rho }{\partial t}+\nabla \cdot \left(\rho \vec{u}\right)=0$$
where the ρ is volume averaged density, $\vec{u}$ is the velocity vector, and t is the time. To incorporate the metallic phase (including both solid and liquid) and the gas phase in the same system, the density was calculated by: (2)$$\rho =\alpha_{m}\rho_{m}+\alpha_{g}\rho_{g}$$
where $\alpha_{m}$ is the metallic-phase fraction, $\alpha_{g}$ is the gas-phase fraction. $\rho_{g}$ is the density of the gas phase. $\rho_{m}$ is the density of the metallic phase, calculated as:(3)$$\rho_{m}=\gamma_{l}\rho_{l}+(1-\gamma_{l})\rho_{s}$$
where $\rho_{l}$ and $\rho_{s}$ are the densities of liquid and solid, respectively. $\gamma_{l}$ is the volume fractions of the liquid phase, updated as a function of the temperature based on the linear interpretation: (4)$$\gamma_{l}=\left\{\begin{array}{cc} 0 & T\leq T_{s}\\ \frac{T-T_{s}}{T_{l}-T_{s}} & T_{s}<T<T_{s}\\ 1 & T\geq T_{L} \end{array}\right.$$
where T is the local temperature, $T_{l}$ and $T_{s}$ are the liquidus and solidus temperatures of the stainless steel 316L, respectively. 

The conservation of momentum is described below with the essential source terms that drive the fluid flow in the melt pool:(5)$$\frac{\partial }{\partial t}\left(\rho \vec{u}\right)+\nabla \cdot \left(\rho \vec{u}\vec{u}\right)=-\nabla p+\nabla \cdot \left(\mu_{eff}(\nabla \vec{u}+\nabla {\vec{u}}^{T})\right)+\rho \vec{g}+{\vec{F}}_{source}$$
(6)$$\mu_{eff}=\alpha_{m}\mu_{m}+\alpha_{g}\mu_{g}$$
(7)$${\vec{F}}_{source}={\vec{F}}_{surface\;tension}+{\vec{F}}_{marangoni}+{\vec{F}}_{recoil\;pressure}+{\vec{F}}_{damping}$$
where p is the static pressure, $\vec{g}$ is the gravity vector, ${\vec{F}}_{source}$ is the momentum source terms, $\mu_{eff}$ is the efficient dynamic viscosity, $\mu_{m}$ and $\mu_{g}$ are the dynamic viscosities of the metallic phase and gas phase, respectively, ${\vec{F}}_{surface\;tension}$, ${\vec{F}}_{marangoni}$, ${\vec{F}}_{recoil\;pressure}$, and ${\vec{F}}_{damping}$ are source terms of surface tension, Marangoni force, recoil pressure [16,58], and momentum loss in semisolid zone. Each term takes the form:(8)$${\vec{F}}_{surface\;tension}=\sigma \kappa \vec{n}\left|\nabla \alpha_{m}\right|\frac{2\rho }{\rho_{m}+\rho_{g}}$$
(9)$${\vec{F}}_{marangoni}=\left[\nabla \sigma -\left(\vec{n}\cdot \nabla \sigma \right)\vec{n}\right]\vec{n}\left|\nabla \alpha_{m}\right|\frac{2\rho }{\rho_{m}+\rho_{g}}$$
(10)$${\vec{F}}_{recoil}=A_{recoil}P_{0}\left\{exp\left[\frac{L_{v}M\left(T-T_{v}\right)}{RTT_{v}}\right]-1\right\}\vec{n}\left|\nabla \alpha_{m}\right|\frac{2\rho }{\rho_{m}+\rho_{g}}$$
(11)$${\vec{F}}_{damping}=\frac{{\left(1-\gamma_{l}\right)}^{2}}{\left(\gamma_{l}^{3}+0.001\right)}A_{mushy}\left(\vec{u}-V_{laser}\right)$$
where σ is the surface tension coefficient. κ is the interface curvature, defined by $\kappa =\nabla \cdot \vec{n}$, where $\vec{n}=\frac{\nabla \alpha_{m}}{\left|\nabla \alpha_{m}\right|}$ is the interface normal factor. $A_{recoil}$ is the recoil pressure coefficient, $P_{0}$ is the ambient pressure, $L_{v}$ is the latent heat of evaporation, $T_{v}$ is the evaporation pressure, M is the molar mass, R is the universal gas constant, $V_{laser}$ is the laser moving velocity, $A_{mushy}$ is the mushy zone constant, which measures the amplitude of the momentum damping in the mushy zone. Note that the value of $A_{mushy}$ has been found in the literature ranging between 1 ×10^3^ and 1 × 10^15^ in the numerical solidification calculations [59]. The larger the value is, the higher the damping becomes, and the faster the velocity drops to zero in the mushy zone [58]. It was utilized between 1 × 10^6^ and 1 × 10^12^ based on factors, including dendrite coherency point, alloy compositions, experiment validation, calculation convergency, and time consumption in most LPBF studies [16,46,60,61]. In this study, the value of 1 × 10^9^ was selected to accommodate the small size of the mushy zone and increase the stability of the numerical solution. 

The free surface of the liquid phase is captured by using the volume of fluid (VOF) algorithm. The conservation of volume fraction is described by:(12)$$\frac{\partial \alpha_{m}}{\partial t}+\nabla \cdot \left(\alpha_{m}\vec{u}\right)=0$$

The standard k-ε turbulence model is applied. The solver based on the SIMPLE algorithm was selected in Fluent to solve the Navier–Stokes equations. 

For the heat transfer and solidification, the equation of conservation of energy is solved:(13)$$\frac{\partial }{\partial t}\left(\rho H\right)+\nabla \cdot \left(\rho \vec{u}H\right)=\nabla \cdot \left(k_{eff}\nabla T\right)+q_{source}$$
where H is the entropy, updated by the phase fraction of the metal and gas:(14)$$H=\alpha_{m}H_{m}+\alpha_{g}H_{g}$$
where $H_{m}$ and $H_{g}$ are the entropies of the metallic phase and gas phase, respectively. They are calculated by:(15)$$H_{g}=\int_{0}^{T}C_{p,g}dT$$
(16)$$H_{m}=\int_{0}^{T}C_{p,m}dT+\gamma_{l}L_{m}$$
where $C_{p,g}$ and $C_{p,m}$ are the specific heats of the gas phase and metallic phase, respectively. $L_{m}$ is the latent heat of melting. 

In Equation (13), $k_{eff}$ is the efficient thermal conductivity, calculated by:(17)$$k_{eff}=\alpha_{m}k_{m}+\alpha_{g}k_{g}$$
where $k_{m}$ and $k_{g}$ are the thermal conductivities of the metallic phase and gas phase, respectively. Similar to the density of the metallic phase, the $k_{m}$ is updated by: (18)$$k_{m}=\gamma_{l}k_{l}+(1-\gamma_{l})k_{s}$$
where $k_{l}$ and $k_{s}$ are the thermal conductivities of the liquid phase and solid phase, respectively.

The energy source terms [16], $q_{source}$, account for evaporation ($q_{evap}$), radiation ($q_{radiation}$), and laser energy input ($q_{laser}$)
(19)$$q_{evap}=-A_{evap}L_{v}\sqrt{\frac{M}{2\pi RT}}P_{0}exp\left[\frac{L_{v}M\left(T-T_{v}\right)}{RTT_{v}}\right]\left|\nabla \alpha_{m}\right|\frac{2\rho }{\rho_{m}+\rho_{g}}$$
(20)$$q_{radiation}=-\sigma_{s}\varepsilon \left(T^{4}-T_{0}^{4}\right)\left|\nabla \alpha_{m}\right|\frac{2\rho }{\rho_{m}+\rho_{g}}$$
(21)$$q_{laser}=\left\{\begin{array}{cc} \alpha_{laser}\frac{0.5\cdot P_{laser}}{\pi r_{laser}^{2}}\frac{\exp\left[-{\left(\frac{r}{2\cdot r_{laser}}\right)}^{2}\right]\left|\nabla \alpha_{m}\right|\frac{2\rho }{\rho_{m}+\rho_{g}}}{\int_{0}^{r_{laser}}2\pi r\exp\left[-{\left(\frac{r}{2\cdot r_{laser}}\right)}^{2}\right]dr} & r\leq r_{laser}\\ 0 & r>r_{laser} \end{array}\right.$$
where $A_{evap}$ is the coefficient of evaporation energy loss, $\sigma_{s}$ is the Stefan–Boltzmann constant, ε is the emissivity, $T_{0}$ is the ambient temperature, $\alpha_{laser}$ is the laser absorptivity, $r_{laser}$ is the laser spot radius, r is the distance from the mesh cell to the laser axis. Note that the mass loss due to evaporation is neglected in the model. 

#### 2.2.2. Simulation Setup and Material Properties 

In order to validate the modeling effectiveness, a three-dimensional symmetrical domain with an overall size of 1500 × 250 × 637.5 μm^3^ and a core size of 1500 × 90 × 270 μm^3^ was created to carry out the simulations for the single-track LPBF process. The domain was meshed into orthogonal grids with sizes from 3.5 µm to 10 µm. The geometry illustration and boundary conditions are shown in Figure 1a. A three-dimensional domain with an overall size of 2900 × 800 × 530 μm^3^, with a mesh size 8 µm, was built for the simulation of multiple-track scanning on a bare plate, as shown in Figure 1b.

Temperature-dependent material properties and other modeling parameters used in the model are shown in Table 2 [15]. Note that the density of the liquid, the specific heat of the solid, and thermal conductivity were approximated as a linear function of the local temperature. Surface tension, which plays a significant role in melt pool instability, has a more complex relationship with the temperature, as shown separately in Figure 2. 

## 3. Results and Discussion

### 3.1. Melt Pool Surface Topography and Dimensions in the Single-Track Experiments

Figure 3 shows the surface topography colormap for single-track samples with and without powder subjected to varying laser powers and speeds, as detailed in Table 1. The top surface of bare plate samples, under the same laser power (260 W) with low laser speeds (0.52 and 0.87 m/s), exhibits flatness and continuity, as shown in Figure 3(a1,b1). No obvious height fluctuation is observed at different locations on the melt pool top surface. However, at a speed of 1.3 m/s (Figure 3(c1)), continuous regions with a higher height (shown as red) occur in the middle of the melt pool. Simultaneously, the lower-height regions (shown as blue) emerge at the boundary of the melt pool. Such melt pool characteristics, where the melt pool has a larger height at the center and a lower height at the melt pool boundaries, are herein termed “swell-undercut”. Upon further increasing the laser speed to over 1.47 m/s, a discontinuous higher region, appearing as balling [62], can be found in Figure 3(d1,e1). With the scanning speed increasing, the morphology of the single tracks on the powder plate shows a similar behavior, transitioning from a smooth top surface at scanning speeds of 0.52 m/s and 0.87 m/s (Figure 3(a2,b2)) to a continuous swell at the middle of the melt pool at 1.3 m/s (Figure 3(c2)), and eventually exhibiting balling for laser speeds exceeding 1.47 m/s (Figure 3(d2)). Moreover, spatters are observed near the melt pool track on the powder plate with a speed higher than 2.2 m/s, as shown in Figure 3(e2,h2). Note that the spatters are evident by the relatively large-sized particles that are not spatially connected to the melt track. 

Comparing the height of each sample on the bare plate or powder plate, the powder plate melt track maintains a higher height under the same process conditions. In contrast to the smooth and regular surface on a bare plate, the presence of an additional layer of powder enhances the overall laser energy absorptivity due to the increased heating surface area and laser reflections among powders [46]. This, in turn, results in more mass being melted into the melt pool, leading to an increase in melt liquid volume and a higher melt pool height.

Under the same laser linear energy density of 300 J/m, the top surface of bare plate single-track samples exhibits increasing instability with increases in both laser power and speed, as shown in Figure 3(b1,f1,g1,h1). The top surface of these samples exhibits features such as swell-undercut, balling, and humping [16], resulting in a poor top surface regularity. Furthermore, when compared to scanning on a bare plate, samples subjected to scanning on a powder plate reveal additional features, including pronounced balling, an exposed melt pool bottom, and spatter escaping the melt pool, as shown in Figure 3(f2,g2,h2), respectively. Similarly, under the same conditions, samples on the powder plate consistently exhibit larger height values than their bare plate counterparts. Both the top surface of single-track samples on a bare plate and a powder plate, with a constant laser speed of 1.47 m/s, display significant swell-undercut and balling, as shown in Figure 3(d1,d2,f1,f2,i1,i2). Notably, the height of balling in the powder plate sample exceeds that in the bare plate sample, as shown in Figure 3(i1,i2), for example. 

Figure 4 presents a cross-sectional view of melt pool morphology of single-track samples on a bare plate and powder plate, corresponding to Figure 3. When maintaining a constant laser power of 260 W, the melt pool for both the bare plate and powder plate transitions from a keyholing mode to a conduction mode as the laser speed increases from 0.52 m/s to 2.20 m/s, shown in Figure 4(a1–e1,a2–e2). This transition is judged by the increase in the width-to-depth ratio, where the start of keyholing has typically been identified when the ratio is less than 1 [12,13,14,20,63]. This transition occurs because the decreasing laser energy deposition is insufficient to sustain the laser penetration [11,12,13,19,61]. Nevertheless, swell-undercut starts forming when the laser speed surpasses 1.30 m/s.

Variations of the melt pool behaviors are observed under the same laser energy density of 300 J/m. Under low laser power and speed conditions, there is sufficient time for heat transfer, resulting in both the bare plate and powder plate melt pools adopting a conduction mode (Figure 4(b1,b2)). As laser power and speed increase, even though the same energy density is maintained, the reduced time for heat transfer intensifies the maximum temperature and recoil pressure. Consequently, keyholing forms, as shown in Figure 4(f1,f2,g1,g2). In addition, swell-undercut phenomena become more pronounced. However, when the laser power and speed exceed 800 W and 2.67 m/s, the keyholing becomes unsustainable, as shown in Figure 4(h1,h2). The extremely strong recoil pressure generated by extremely high power and high speed forces the liquid away from the melt pool. This expulsion restricts the energy deposited into the melt pool and limits the melt pool from going deeper. From Figure 4(d1,f1,i1) and Figure 4(d2,f2,i2), the melt pool morphology of both the bare plate and powder plate samples, with a constant laser speed of 1.47m/s, transfers from the conduction mode to the keyholing mode as the laser power increases from 260 W to 620 W. The transformation to keyholing initiates when the laser power exceeds 440 W, indicating that an increasing power level results in greater energy deposition from the laser. Importantly, all cases exhibit the presence of the swell-undercut defect when subjected to a laser speed of 1.47 m/s.

Figure 5, Figure 6 and Figure 7 present analyses of melt pool width, depth, and width-to-depth ratio under consistent laser power (260 W), constant laser energy density (300 J/m), and unvarying laser speed (1.47 m/s), respectively. Both melt pool width and depth exhibit a decreasing trend with escalating laser speed for both the powder plate and bare plate samples, while the width-to-depth ratio increases, as shown in Figure 5. The increase in laser speed corresponds to a reduction in energy density, leading to a decrease in the size of the melt pool (both width and depth). Notably, at a laser speed of 0.52 m/s, where the energy density is sufficient to sustain keyholing, the width-to-depth ratio is particularly low at 0.633. In all other cases with the same laser power, the width-to-depth ratio exceeds 1. 

From Figure 6, the melt pool width and width-to-depth ratio initially decrease and then increase with increasing laser power for both the bare plate and powder plate samples under a constant laser energy density of 300 J/m. The melt pool depth follows an opposite trend, first increasing and then decreasing. As discussed earlier, the melt pool transitions from a conduction mode to keyholing mode as the laser power and speed increase from 260 W and 0.87 m/s to 620 W and 2.07 m/s, accompanied by a decrease in the width-to-depth ratio to above 1.25. However, the melt pool exits the keyholing mode and goes to an unstable-behaved mode when the laser power and speed reach 800 W and 2.67 m/s, due to insufficient time for heat transfer. The melt pool dimensions of both the bare plate and powder plate change with laser power under the same laser speed of 1.47 m/s, which is relatively monotonous (Figure 7). As the laser power increases from 260 W to 620 W, the melt pool width and depth keep increasing, while the width-to-depth ratio decreases. This trend is attributed to the increased energy deposition and subsequent transition to the keyholing mode.

### 3.2. Numerical Analysis of Single-Track Laser Scanning

#### 3.2.1. Mesh Sensitivity and Model Validation

Considering the potential for manual error in measuring melt pool dimensions in simulations, the analysis for mesh sensitivity involves using the numerically calculated melt pool volume instead. This approach helps mitigate inaccuracies that might arise due to manual measurement errors, ensuring a more reliable assessment of mesh sensitivity.

Figure 8 shows the melt pool volume and computational time under different mesh sizes in the core region for case N06 (laser powder 440 W and laser speed 1.47 m/s). Reducing the mesh size in the core region from 9 µm to 7 µm, and, subsequently, to 5 µm, the melt pool volume changes from 79,040 µm^3^ to 884,020 µm^3^ and, subsequently, to 900,573 µm^3^. The melt pool volume variations represent approximately 12% and 2%, respectively, in two mesh size reductions. However, computational times increased notably, reaching 6.8 h, 13.9 h, and 35.6 h as the mesh size decreased from 9 µm to 7 µm and then to 5 µm. The computational time variations are significant, showing increments of 104% and 156%, respectively. Additionally, the convergence criterion for computational results is defined as a melt pool volume variation within 0.1%, calculated at intervals equal to the laser diameter divided by the laser speed.

To balance the computational costs and accuracy, a mesh size of 7 µm was employed in this study for all single-track simulations. This selection ensures the reliability of the results, while optimizing computational efficiency.

To understand surface formation under different working parameters, four cases were chosen for simulation to capture the melt pool dynamics in the LPBF process. Table 3 shows the selected cases and the comparison of melt pool dimensions between experimental data and simulation results. Note that the direct comparison of the predicted melt pool morphologies is shown in Figure 9, which will be discussed in detail next. All cases exhibit errors within a 10% range, except for the case N05-P260V2.20. As discussed in Section 2.1, cases with different laser powers and speeds use the same modeling assumptions, especially laser absorptivity and temperature-dependent recoil pressure models, for simplification. However, it is more complicated in real LPBF processing. The impact of vapor plumes, a dynamic melt pool free surface at the scanning spot, and pressure fluctuation with different laser powers and speeds varies on the energy deposition and recoil pressure. Furthermore, the goal of this paper is to figure out the basic top surface formation mechanism. In this way, the errors between the experiment and simulation, as shown in Table 3, are acceptable.

#### 3.2.2. Physical Mechanisms of Swell-Undercut Formation

Figure 9 shows the simulation and experiment results of single-track laser scanning on the bare plate under the selected laser powers and speeds listed in Table 3. The simulation consistently reproduces the melt pool size on the cross section, exhibiting notable agreement with the experimental data. This agreement is particularly evident in capturing the swell-undercut defect, as indicated by the comparisons in Figure 9(a2) vs. Figure 9(a3), Figure 9(b2) vs. Figure 9(b3), Figure 9(c2) vs. Figure 9(c3), and Figure 9(d2) vs. Figure 9(d3). In all cases, material undergoes melting at the scanning spot and flows downward with a maximum velocity below 10 m/s. However, the dynamics of the melt pool vary under different working parameters, as shown in Figure 9(a1–d1).

Lower laser speed (e.g., 0.52 m/s) allows for increased energy deposition, resulting in the formation of a short and deep melt pool exhibiting a keyholing mode. This characteristic supports the liquid sufficiently, enabling it to flow back and refill the voids caused by recoil pressure. Consequently, a flat surface is formed before solidification, as illustrated in Figure 9(a1,a2). However, as the laser speed increases from 0.52 m/s to 2.20 m/s under a constant laser power of 260 W, the melt pool becomes shallower and narrower due to decreased energy density. 

Comparing the lower power case N04-P260V1.47 (Figure 9(b1–b3)) with the higher power case N06-P440V1.47 (Figure 9(d1–d3)), it is observed that the higher power case exhibits a longer and deeper melt pool and the undercut depth increases. This suggests that increased energy deposition by increasing laser power, while keeping the identical laser speed, worsens the swell-undercut defect, when such a defect is already present. Higher power increases energy deposition, intensifying evaporation and recoil pressure at the free surface of the scanning spot. The enhanced pressure can further squeeze liquid downward and backward to the melt pool end more dramatically, thereby promoting the formation of the swell-undercut defect, as evident in the comparison between Figure 9(b2,d2). However, increased energy deposition by reducing the laser speed, instead, relieves the swell-undercut defect, as evidenced by comparing Figure 9(a1,a2,b1,b2). The low speed allows more time for heat transfer through the material and for the liquid to flow back and refill the voids caused by recoil pressure before the liquid solidifies. Nevertheless, the undercut appears when the laser speed exceeds 1.47 m/s, as shown in Figure 9(a2–d2). Overall, the formation of swell-undercut during laser scanning is driven by intense recoil pressure and insufficient liquid refilling. This is attributed to the fact that, with sufficient energy deposition together with high laser speed conditions, the liquid solidifies rapidly before it has sufficient time to flow back and refill the void caused by recoil pressure. 

To assist in understanding swell-undercut formation, the simulation results of case N04-P260V1.47 are illustrated in detail, with different cross sections showing the liquid fraction and temperature in Figure 10. As the material melts and evaporates at the scanning spot, the liquid experiences significant force from the intense recoil pressure, flowing towards the melt pool end (Figure 10(c1–c6)) and exhibiting high velocity under the laser spot (Figure 10(b1)). Once the liquid flows away from the scanning spot, the fluid velocity drops rapidly from approximately 10 m/s to 1 m/s, as the semisolid zone damps kinetic energy, as shown in Figure 10(b2–b6). Nevertheless, the liquid solidifies before refilling the volume of the void created by recoil pressure. This occurs because the laser moves forward too rapidly under high laser speed conditions, leaving insufficient energy for the liquid to refill the void (Figure 10(a2–a4)). In contrast, lower laser power and speed conditions (260 W and 0.52 m/s) provide more time and energy deposition for liquid refilling, resulting in a different outcome, shown in Figure 9(a1,a2). As a result, the melt pool swell-undercut forms, which significantly increases the top surface roughness.

### 3.3. Surface Topography of Cubic Samples and Multiple-Track Simulation

The top surface topography of cube samples and the simulation of multiple-track scanning on the bare plate are briefly discussed in this section to present connections between the melt pool characteristics and the top surface roughness. 

Figure 11 shows the top surface topography of cubic samples and multiple-track simulation for abare plate conducted under two distinct laser processing conditions: high laser power and speed case (440 W and 1.47 m/s) and low laser power and speed case (260 W and 0.52 m/s). Elevated laser power and speed result in high recoil pressure and reduced heat transfer time, promoting the swell-undercut, as shown in Figure 11(a1). Such a feature creates poor surface roughness on the top surface due to the large height fluctuation. In contrast, the top surface, under low laser power and speed conditions (Figure 11(b1)) presents enhanced smoothness and reduced height fluctuations, with only a few partially melted particles. This outcome is owing to the adequate heat transfer time and energy deposition. Notably, the observed top surface characteristics are aligned with findings in the single-track experiments, as discussed in Section 3.1. 

The numerical modeling for the single-track laser scanning process was employed here to simulate multiple tracks, mimicking the scanning process on the top surface. Figure 11(a2) shows a similar swell-undercut feature for the high-power and high-speed case, where distinguishable fluctuations of height can be observed across the laser scanning tracks. In addition, a large balling volume at the laser start point (Figure 11(a2), red region on the left) and a large volume of voids at the laser end point (Figure 11(a2), blue region on the right) were formed due to the large recoil pressure and fast solidification speed. This phenomenon is commonly observed during the initiation and ending stages of the laser scanning process. With the low-power and low-speed conditions, a smooth surface was predicted with minimal balling and voids during the laser starting and ending stages (Figure 11(b2). Although the absolute height differences vary between experiments and simulations, the key features of the melt pool behaviors and the surface topography have been captured by the computational model. 

This demonstration highlights the versatility of the laser–matter interaction model, showcasing its applicability in predicting melt pool behaviors and potential defects during the fabrication process. Despite the current model not incorporating powder, it proves valuable in offering insights that aid in elucidating the forming mechanisms of specific features or defects of interest. By leveraging the model, researchers can gain a deeper understanding of the intricate dynamics involved in laser-based manufacturing processes, contributing to improved process control and the development of more robust fabrication techniques. 

## 4. Conclusions

This study employed single-track laser scanning experiments, both with and without a powder layer, to investigate key topographic features during laser–matter interaction. The experiments revealed a transition from a conduction mode to a keyholing model and identified phenomena such as balling, humping, swell-undercut, and spattering across a broad range of process conditions. A high-fidelity computational model integrating fluid dynamics, heat transfer, vaporization, and solidification was developed to interpret the experimental results and elucidate melt pool characteristics in detail. The key conclusions derived from both the experiments and simulations are summarized as follows:At a low power of 260 W, the width and depth of the melt pool decrease with increasing laser speed in single-track scanning, both with and without a powder layer. Swell-undercut appears when the speed exceeds 1.30 m/s. At a higher power and a laser speed of 1.47 m/s, the melt pool expands, resulting in significant balling.Increasing laser power and speed at a linear energy density of 300 J/m leads to an increase in melt pool width and depth, transitioning from shallow and wide to deep and narrow and then back to a shallow and wide melt pool. Balling and swell-undercut phenomena emerge beyond 440 W and 1.47 m/s, with additional spattering observed beyond 620 W and 2.67 m/s.The laser–matter interaction on the bare plate or powder plate does not alter the melt pool characteristics. However, the fluctuation of melt track height on a powder plate is consistently higher than on a bare plate in the laser moving direction, due to increased laser energy absorptivity and additional mass from loose powder, worsening melt pool irregularity.The formation of swell-undercut, revealed by the numerical model, is owing to the large void space created by high intense recoil pressure and insufficient liquid refilling due to momentum damping in the semisolid zone and rapid solidification at high laser power and high-speed conditions. In contrast, a lower laser power and speed provide adequate heat transfer time and energy deposition, reducing swell-undercut and surface roughness.Cubic samples under high laser power and speed exhibit swell-undercut and increased height fluctuation, while lower laser power and speed result in smoother surfaces due to enhanced heat transfer and energy deposition. These characteristics align with findings from multiple-track scanning simulations, highlighting the influence of insufficient energy deposition and reduced heat transfer time on surface topography.

The accurate prediction of melt pool dynamics in LPBF is pivotal for advancing both materials’ and process development, targeting the desired surface finishing, microstructure, and, ultimately, material properties. The current study employs the established framework in Ansys Fluent, captures the essential physics that drives melt pool formation, and elucidates surface characteristics. With the introduction of the melt pool modules in commercial software packages, such capabilities have become increasingly accessible for both scientific research and engineering applications. Indeed, ongoing development will continue to refine the underlying assumptions and governing physics in the models for a more accurate representation of the intricate dynamics in LPBF. However, such models are still limited to a countable number of tracks and layers because of the required high spatial and temporal resolutions and the resultant prolonged computational time. Beyond adapting advanced numerical algorithms and utilizing high-performance computing facilities, a promising direction involves a novel scheme that solves not only the melt pool dynamics locally but also the global thermal history within complex geometries. The advent of artificial intelligence presents a transformative opportunity to expedite model setup, computation, and post-processing through data analytics. This technological leap is anticipated to significantly enhance the efficiency of LPBF simulations. Furthermore, by integrating with other mechanistic models predicting microstructure and material properties, the synergy between simulations and LPBF research not only fosters innovations in AM but also accelerates its broader success in industrial applications.

## Figures and Tables

**Figure 1 micromachines-15-00170-f001:**
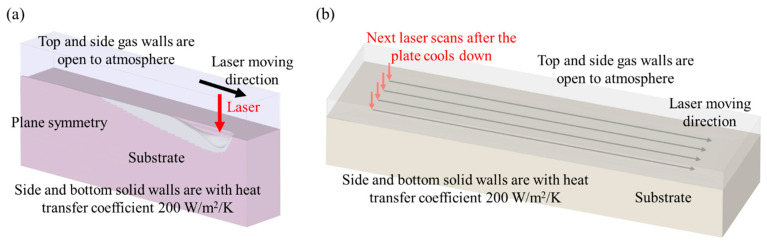
Boundary conditions: (**a**) single-track, (**b**) multiple-track.

**Figure 2 micromachines-15-00170-f002:**
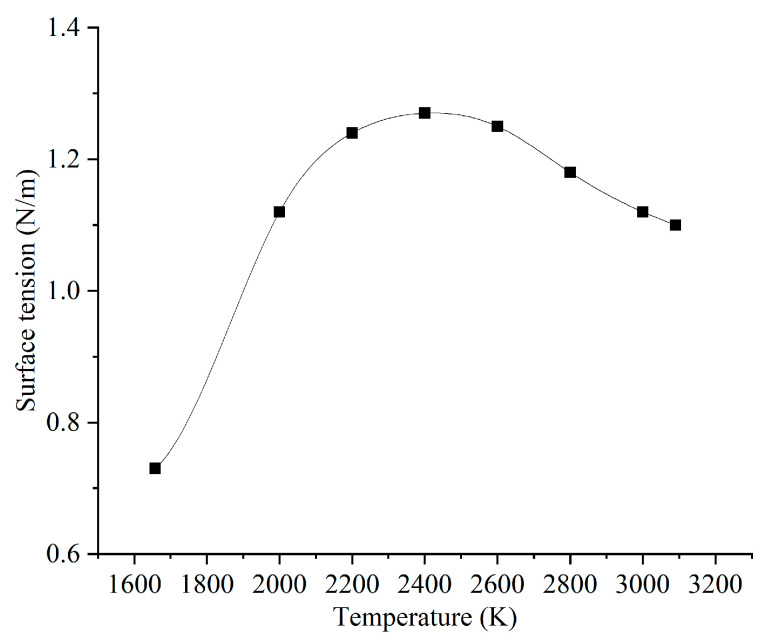
Surface tension [16].

**Figure 3 micromachines-15-00170-f003:**
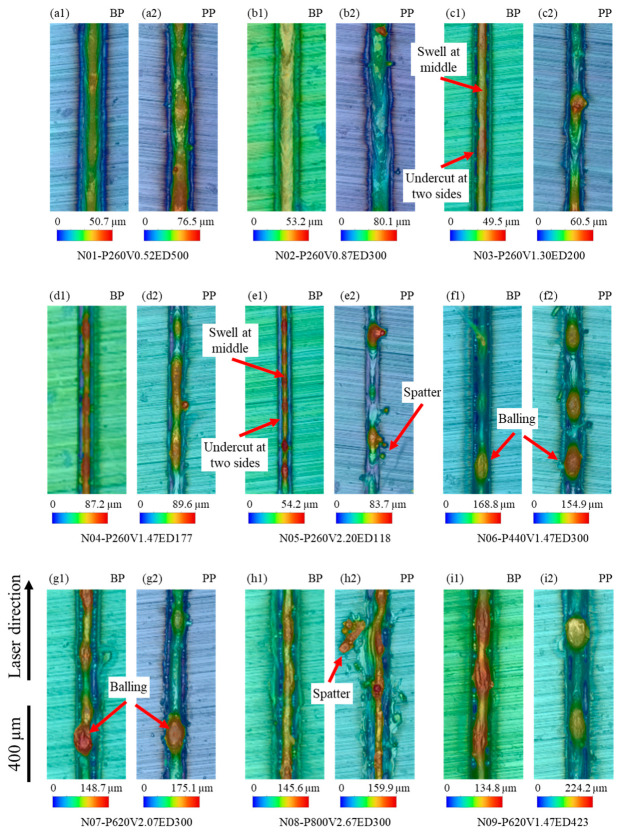
Surface topography colored by height for single-track samples on bare plate (BP) and powder plate (PP) with different laser powers and speeds. (**a1**,**a2**) P = 260 W, V = 0.52 m/s; (**b1**,**b2**) P = 260 W, V = 0.87 m/s; (**c1**,**c2**) P = 260W, V = 1.30m/s; (**d1**,**d2**) P = 260 W, V = 1.47 m/s; (**e1**,**e2**) P = 260 W, V = 2.20 m/s; (**f1**,**f2**) P = 440 W, V = 1.47 m/s; (**g1**,**g2**) P = 620 W, V = 2.07 m/s; (**h1**,**h2**) P = 800 W, V = 2.67 m/s; (**i1**,**i2**) P = 620 W, V = 1.47 m/s.

**Figure 4 micromachines-15-00170-f004:**
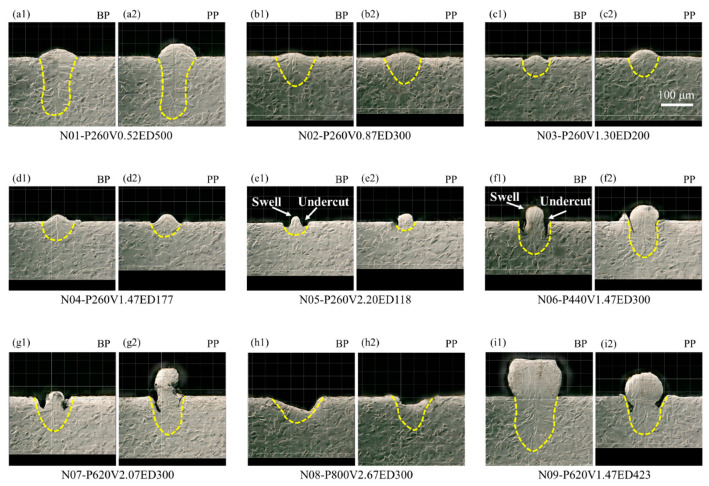
Cross-sectional view of melt pool morphology on bare plate (BP) and powder plate (PP) with different laser powers and speeds. (**a1**,**a2**) P = 260 W, V = 0.52 m/s; (**b1**,**b2**) P = 260 W, V = 0.87 m/s; (**c1**,**c2**) P = 260W, V = 1.30m/s; (**d1**,**d2**) P = 260 W, V = 1.47 m/s; (**e1**,**e2**) P = 260 W, V = 2.20 m/s; (**f1**,**f2**) P = 440 W, V = 1.47 m/s; (**g1**,**g2**) P = 620 W, V = 2.07 m/s; (**h1**,**h2**) P = 800 W, V = 2.67 m/s; (**i1**,**i2**) P = 620 W, V = 1.47 m/s.

**Figure 5 micromachines-15-00170-f005:**
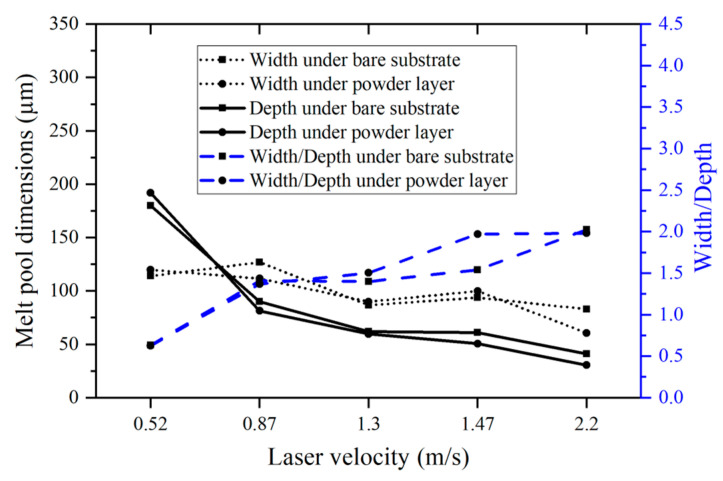
Melt pool width, depth, and width-to-depth measured from the printed single-track samples under the same power (260 W).

**Figure 6 micromachines-15-00170-f006:**
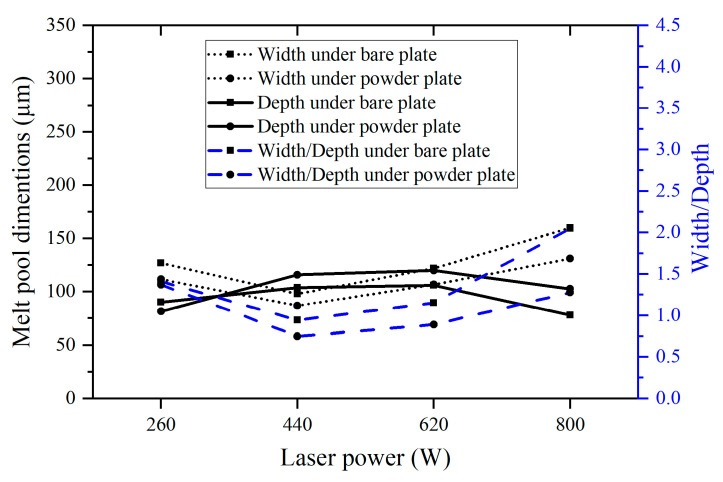
Melt pool width, depth, and width/depth measured from the printed single-track samples under the same laser energy density (300 J/m).

**Figure 7 micromachines-15-00170-f007:**
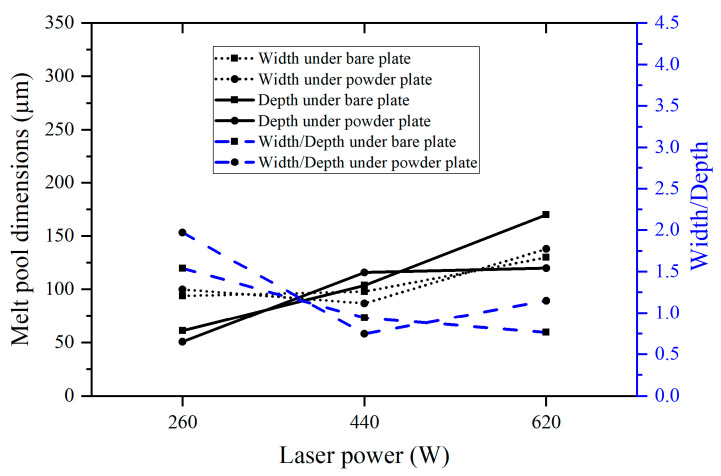
Melt pool width, depth, and width/depth measured from the printed single-track samples under the same laser speed (1.47 m/s).

**Figure 8 micromachines-15-00170-f008:**
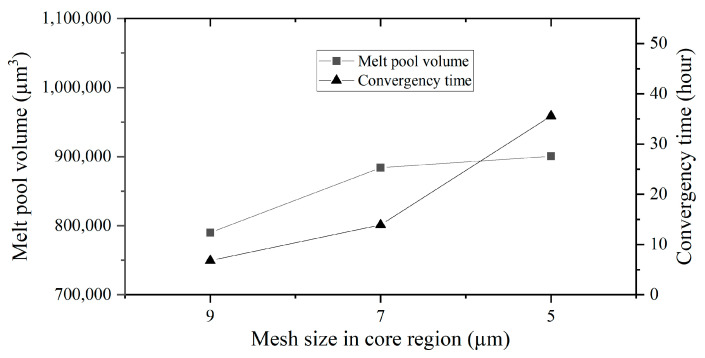
Melt pool volume and computational time under different mesh sizes in the core region.

**Figure 9 micromachines-15-00170-f009:**
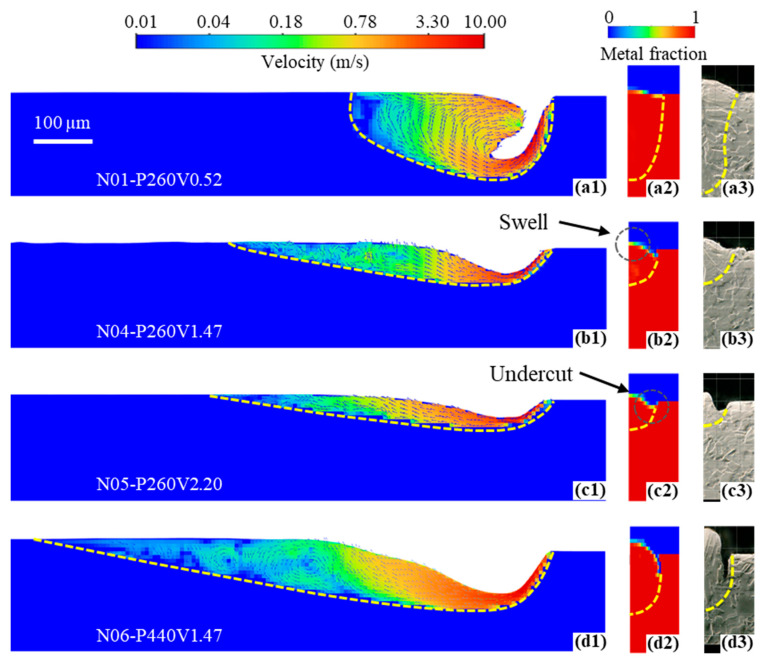
Predicted velocity and melt pool in simulations (**a1**–**d2**) and melt pool cross sections in experiments (**a3**–**d3**) under different laser powers and speeds. (**a1**–**a3**) P = 260 W, V = 0.52 m/s; (**b1**–**b3**) P = 260 W, V = 1.47 m/s; (**c1**–**c3**) P = 260 W, V = 2.20 m/s; (**d1**–**d3**) P = 440 W, V = 1.47 m/s.

**Figure 10 micromachines-15-00170-f010:**
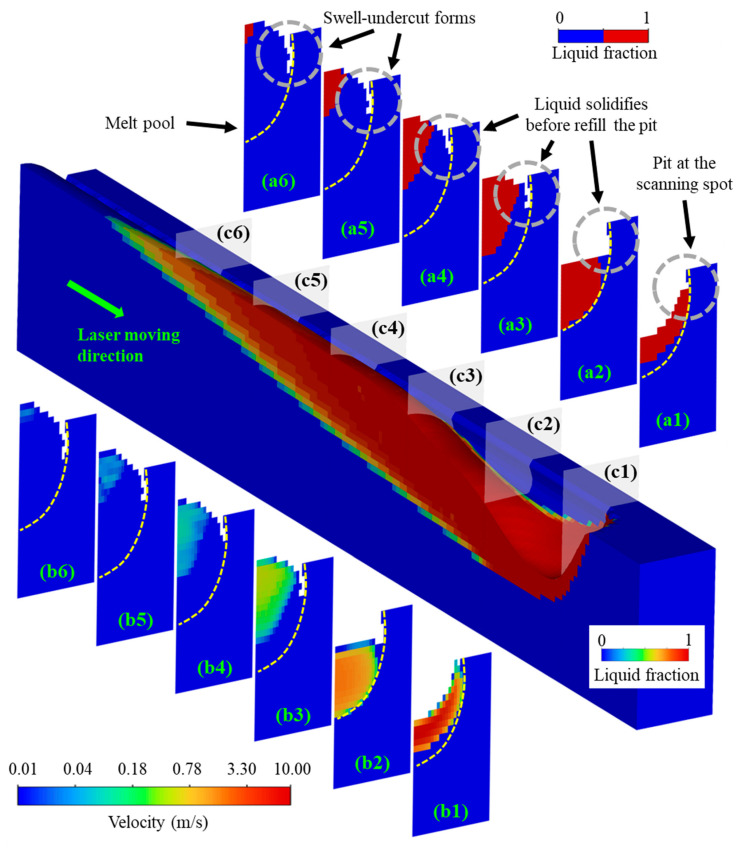
Melt pool swell-undercut formation with laser power 440 W and speed 1.47 m/s, (**a1**–**a6**), liquid fraction, (**b1**–**b6**), velocity, (**c1**–**c6**), location of cross sections.

**Figure 11 micromachines-15-00170-f011:**
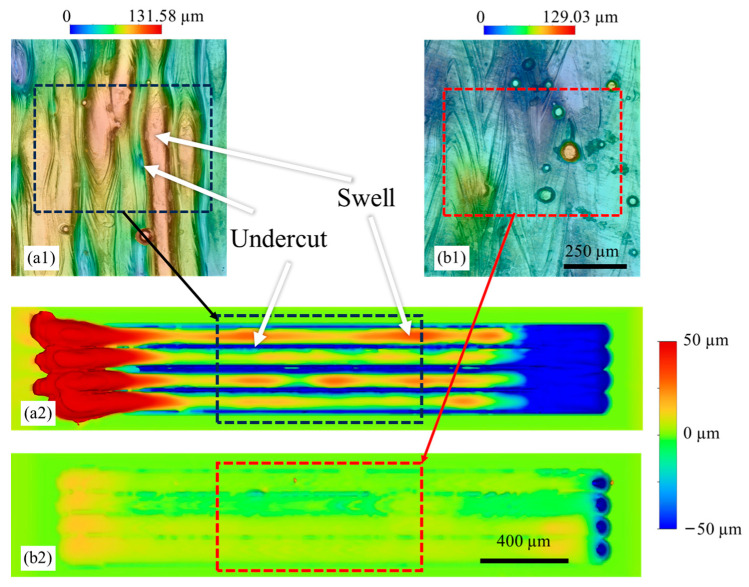
Surface topography of cube samples and multiple-track simulation, (**a1**) experiment and (**a2**) simulation for the high laser power and speed condition (440 W and 1.47 m/s); (**b1**) experiment and (**b2**) simulation for the low laser power and speed condition (260 W and 0.52 m/s).

**Table 1 micromachines-15-00170-t001:** Experimental parameters for single tracks.

Case Name	Laser Power, W	Laser Speed, m/s	Laser Energy Density, J/m
N01	260	0.52	500
N02	260	0.87	300
N03	260	1.30	200
N04	260	1.47	177
N05	260	2.20	118
N06	440	1.47	300
N07	620	2.07	300
N08	800	2.67	300
N09	620	1.47	423

**Table 2 micromachines-15-00170-t002:** Material properties and model parameters [16].

Properties	Symbol	Value	Unit
Density of solid phase	$\rho_{l}$	7950	kg/m^3^
Density of liquid phase	$\rho_{s}$	8200 − 0.77T	kg/m^3^
Density of gas phase	$\rho_{g}$	1.22	kg/m^3^
Specific heat of solid phase	$C_{p,m}$	415 + 0.1838T	J/kg/K
Specific heat of liquid phase	$C_{p,l}$	830	J/kg/K
Specific heat of gas phase	$C_{p,g}$	1006.43	J/kg/K
Thermal conductivity of solid phase	$k_{s}$	9.23 + 0.0139T	W/m/K
Thermal conductivity of liquid phase	$k_{l}$	5.5 + 0.0133T	W/m/K
Thermal conductivity of gas phase	$k_{g}$	0.02	W/m/K
Solidus temperature	$T_{s}$	1658	K
Liquidus temperature	$T_{l}$	1723	K
Evaporation temperature	$T_{v}$	3090	K
Latent heat of melting	$L_{m}$	2.6 × 10^5^	J/kg
Latent heat of vaporization	$L_{v}$	7.45 × 10^6^	J/kg
Viscosity of metallic phase	$\mu_{m}$	0.006	N/m^2^
Viscosity of gas phase	$\mu_{g}$	1.85 × 10^−5^	N/m^2^
Mushy zone constant	$A_{mushy}$	1 × 10^9^	kg/m^3^/s
Molar mass	M	0.05593	kg/mol
Ambient pressure	$P_{0}$	1.013 × 10^5^	Pa
Universal gas constant	R	8.314	kgm^2^/s^2^/K/mol
Stefan–Boltzmann constant	$\sigma_{s}$	5.67 × 10^−8^	kg/s^3^/K^4^
Emissivity	ε	0.5	
Coefficient of evaporation energy loss	$A_{evap}$	0.5	
Recoil pressure coefficient	$A_{recoil}$	0.6	
Laser absorptivity	$\alpha_{laser}$	0.5	
Laser spot radius	$r_{laser}$	50	µm

**Table 3 micromachines-15-00170-t003:** Comparison of melt pool dimensions between experiment and simulation.

	Experimental Melt Pool Size on Bare Plate		Numerical Melt Pool Size on Bare Plate		Error of Simulation	
	Width	Depth	Width	Depth	Width	Depth
N01-P260V0.52	114 µm	180 µm	119 µm	168 µm	4.20%	7.14%
N04-P260V1.47	94 µm	61 µm	99 µm	62 µm	5.05%	1.61%
N05-P260V2.20	83 µm	41 µm	92 µm	48.3 µm	9.78%	15.11%
N06-P440V1.47	98 µm	104 µm	105 µm	108 µm	6.67%	3.70%

## Data Availability

Data are contained within the article.

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
