# Peer review of "Understanding Melt Pool Behavior of 316L Stainless Steel in Laser Powder Bed Fusion Additive Manufacturing"

_micromachines, 2024, doi:10.3390/mi15020170_

Round 1

Reviewer 1 Report

Comments and Suggestions for Authors

This manuscript investigates the effect of different process parameters on melt pool characteristics and surface morphology complexity. And a computational model is developed to predict the melt pool patterns. There are a number of issues that need to be addressed before acceptance

1.     What is the significance of Figure 1, or what is Figure 1 trying to illustrate, since the content of Figure 1 is given exactly in Table I.

2.     The formulas for the physical model in Chapter 2 should give a citation.

3.     There seems to be an error in the physical parameters in Table II, e.g., the molar mass of 316L stainless steel should be 0.05593 instead of 1e-6, and the permeability coefficient should be 1e6 instead of 1e9.

4.     In the analysis of Fig. 4, the authors argue that spattering occurs in E2, what is the reasoning behind the belief that it is spattering, and why is it not wetting behavior resulting from the instability of the melt pool.

5.     The scale is missing in Figure 4, and is given in only one figure, not in any of the others.

6.     Line 244, "The presence of an additional layer of powder on top of the bare plate allows loose powder to come into direct contact with the laser, increasing laser absorptivity. increasing laser absorptivity." Meaning that the powder has a higher laser absorptivity?

7.     In Fig. 5 the authors suggest that the melt pool changes from a keyhole mode to a conduction mode as the laser speed increases, but the keyhole does not appear to be visible in the figure.

8.     Line 279, there is a textual error such as (Figure 5 (b1) and (b1)), please recheck

9.     For the analysis of Figs. 6-8, it is suggested to reconsider that there seems to be some problems, for example, for a fixed energy density, the authors found that the width-to-depth ratio of the molten pool shows a first decrease and then an increase with increasing energy density? Is the energy density fixed, and is it the energy density or other process parameters that vary.

10.  The analysis of prediction errors in Table III would be better represented graphically.

11.  In the analysis of Fig. 10, the authors suggest that an increase in recoil pressure promotes the formation of a "swell-undercut defect", however, in the comparison of a1 and c1, the recoil pressure in a1 is significantly higher, but there is no obvious "swell-undercut defect".

Reviewer 2 Report

Comments and Suggestions for Authors

This manuscript investigated the top surface morphology formation behavior with and without a powder layer on the 316L plate. The experiments demonstrated through varing laser parameters such as laser energy and scanning speed diverse morphology including balling, humping and swell-undercut are formed. A numerical model was developed and elucidate the formation behavior in detail. The obtained results are interesting to the broad readership of Micromachines. Before this manuscript becomes acceptable, please carefully address the following comments.

1. In line 71, please add SPACE between “simulated” and “316L”.

2. Please add SPACE between value and unit.

3. In line 150, please replace HNO3 with HNO3.

4. In line 154, please replace L-PBF with LPBF. Or use L-PBF instead in the whole manuscript.

5. Equation layout should be refined.

6. In line 279, the second b1 should be b2.

7. In line 321, J/s should be J/m.

8. The font format of numbers in Fig. 8 and 9 should be consistent with other figures.

9. In section 3.3, does the swell show spatial periodicity along the scanning direction? Does it correlate with scanning speed?

Based on the abovementioned comments, this manuscript is recommended for major revision. A revised manuscript is required.

Round 2

Reviewer 1 Report

Comments and Suggestions for Authors

The authors addressed most of my concerns. Some minor suggestions are as follows, 

(1)Recent advances in laser powder bed fusion additive manufacturing are recommended to add in the introduction, such as: https://doi.org/10.1080/10408436.2022.2041396 and https://doi.org/10.1016/j.ijmachtools.2023.104099

(2)With the introduction of Flow3D and other commercial software for molten pool simulation modules, what is the development direction of research? Could the author look forward to the future academic research direction at the conclusion

Reviewer 2 Report

Comments and Suggestions for Authors

The authors have answered all comments well. The manuscript is recommended for publication.

Author Response

Thank you once more for your review of the paper and for recommending it for publication.